# A 100-m gridded population dataset of China's seventh census using ensemble learning and geospatial big data

Yuehong Chen[1], Congcong Xu[1], Yong Ge[2], Xiaoxiang Zhang[1], and Ya'nan Zhou[1]

[1]College of Geography and Remote Sensing, Hohai University, Nanjing 211100, China
[2]State Key Laboratory of Resources and Environmental Information System, Institute of Geographical Sciences and Natural Resources Research, Chinese Academy of Sciences, Beijing 100101, China

*Correspondence to*: Yuehong Chen and Yong Ge (yuehong@hhu.edu.cn and gey@lreis.ac.cn)

**Abstract.** China has undergone rapid urbanization and internal migration in past years and its up-to-date gridded population
datasets are essential for various applications. Existing datasets for China, however, suffer from either outdatedness or failure to incorporate the latest seventh national population census data conducted in 2020. In this study, we develop a novel population downscaling approach that leverages stacking ensemble learning and geospatial big data to produce up-to-date population grids at a 100-m resolution for China from the seventh census data at both county and town levels. The proposed approach employs stacking ensemble learning to integrate the strengths of random forest, XGBoost, and LightGBM through
fusing their predictions in a training mechanism and it delineates the inhabited areas from geospatial big data to enhance the gridded population estimation. Experimental results demonstrate that the proposed approach exhibits the best fit performance compared to individual base models. Meanwhile, the out-of-sample town-level test set indicates that the estimated gridded population dataset ($R^2$=0.8936) is more accurate than existing WorldPop ($R^2$=0.7427) and LandScan ($R^2$=0.7165) products for China in 2020. Furthermore, with the inhabited areas enhancement, the spatial distribution of population grids is more
reasonable intuitively than the two existing products. Hence, the proposed population downscaling approach provides a valuable option for producing gridded population datasets. The estimated 100-m gridded population dataset of China holds great significance for future applications and it is publicly available at https://figshare.com/s/d9dd5f9bb1a7f4fd3734 (Chen et al., 2024b).

## 1 Introduction

Human population distribution is a critical factor in measuring, mapping and understanding human-nature interactions (Leyk et al., 2019; Wardrop et al., 2018). It serves as a fundamental variable in a wide range of applications (Baynes et al., 2022; Yi et al., 2019), including exposure to disasters and pollutants (Zhang et al., 2022; Fang et al., 2014; Nadim et al., 2006; Macmanus et al., 2021), access to resources and facilities (Song et al., 2018; Chen et al., 2023; Tatem and J., 2014; Linard et al., 2010), and impact on environment (Feng et al., 2021; Zhou et al., 2021; Wang et al., 2020; Samir and Lutz, 2017). The

general manner of collecting population distribution data is through demographical data linked to spatial boundary datasets (e.g., administrative units) (Chen et al., 2019; Leyk et al., 2019). Census data are the primary source of demographic information; however, this form of population data provides only a single value for each irregular census unit and cannot specify detailed distribution at grid scales within each census unit (Wardrop et al., 2018; Qiu et al., 2022). Converting irregular census data into regular population grids (termed as population spatialization or population downscaling) proves to be an effective technique to overcome the limitations of census data. Gridded population data have been widely acknowledged their benefits in integration with other gridded spatial variables, such as remote sensing products (Leyk et al., 2019; Chen et al., 2019).

In recent years, a variety of applications have shown an increasing demand for gridded population data (Kubíček et al., 2018; Stevens et al., 2019; Chen et al., 2024a). Timely and reliable gridded population data are highly desired to meet this demand, especially in countries experiencing rapid urbanization and internal migration like China. In China, informed decision-making and sustainable urban development greatly depend on timely and accurate gridded population distribution data (Chen et al., 2020b; Cheng et al., 2020; Guo et al., 2023b; Tu et al., 2022). The Seventh National Population Census of China, conducted in 2020, presents a valuable opportunity to produce the required up-to-date and reliable gridded population data.

In recent years, continuous and significant efforts have been made to generate several gridded population data for China. Ye et al. (2019) used a random forest algorithm to downscale the Sixth National Population Census data of China in 2010 to gridded population data at a 1 km resolution. Zhao et al. (2020) converted China's county-level population data in 2015 into 1-km gridded population. Cheng et al. (2020) combined the random forest algorithm and area-to-point kriging to disaggregate the town-level population sample survey data of China in 2015 to 1-km monthly population grids. Chen et al. (2022a) employed geographically weighted regression to generate 0.01° population grids from the county-level population of China in 2018. Tu et al. (2022) utilized human digital footprints to produce gridded population dynamics at a 0.01° resolution in 2018. Chen et al. (2020b) leveraged existing gridded population data to simulate future gridded population distribution every five years from 2015 to 2050. Chen et al. (2020c) projected provincial population data from 2010 to 2100 under shared socioeconomic pathways and spatially allocated the projected population into grids at a 30 arc-seconds resolution. Apart from estimating population grids at the national scale in China, recent similar efforts have been made for individual cities (Chen et al., 2019; Guo et al., 2023a; Yang et al., 2023; Liu et al., 2023; Wu et al., 2020; Zhao et al., 2021a; Zhao et al., 2021b) and provinces (Gao et al., 2021; Yi et al., 2019). Additionally, several global gridded population datasets are available for China, including LandScan and WorldPop (Bright and Coleman, 2000; Tatem, 2017). The LandScan program, initiated at Oak Ridge National Laboratory, provides global yearly population grids at a 30 arc-seconds resolution (Bright and Coleman, 2000). The WorldPop, a research project launched in the United Kingdom, also offers global yearly population grids up to 2020 with a higher resolution of 100-m (Tatem, 2017; Stevens et al., 2015). Although these efforts can provide abundant gridded population datasets for China, they are either outdated (several datasets before 2020) or a lack of utilizing the actual county-level and finer town-level Seventh National Population Census data of China in 2020.

In past years, these studies have been developed various methods to downscale population census data to population grids. However, they usually employ a single machine learning method to model the complex relationship between population and its auxiliary variables (i.e., covariates) when producing gridded population datasets (Stevens et al., 2015; Ye et al., 2019; Zhao et al., 2020; Bright and Coleman, 2000). Individual machine learning methods often have their inherent disadvantages (e.g., overfitting and instability), which can be addressed by a recently popular way of ensemble learning to simultaneously take advantages of multiple homogeneous and heterogeneous individual methods (Yao et al., 2022; Tu et al., 2022; Costache and Bui, 2019; Fang et al., 2021). Ensemble learning is a technique in machine learning and focuses on combining multiple algorithms to improve the overall performance and robustness of predictions (Dong et al., 2020). Stacking learning is considered to be one of the most effective ensemble learning techniques due to its utilization of a training mechanism to merge the predictions of individual machine learning algorithms (Costache and Bui, 2019). Ensemble learning has been widely recognized its merits in various applications (Dong et al., 2020; Wu et al., 2021; Xu et al., 2023). For instance, Yao et al. (2022) demonstrated the stacking ensemble learning outperformed typical individual machine learning algorithms in evaluating flash flood susceptibility. Despite its success in other domains, the potential of stacking ensemble learning in population disaggregation remains relatively unexplored.

To address these research and data gaps in the current literatures, we develop a novel population downscaling approach that leverages stacking ensemble learning and geospatial big data to generate a 100 m gridded population dataset for China from the seventh census data in 2020. The county-level and town-level census data of China in 2020 and ten related covariates at the 100-m resolution were first collected as the input datasets. Subsequently, three popular machine learning algorithms (i.e., random forest, XGBoost, and LightGBM) were chosen as base models to create and train the stacking ensemble learning to generate gridded population dataset for China. Finally, we assessed the generated gridded population dataset using the town-level census data and compared it with the Landscan and WorldPop datasets.

## 2 Data

Three types of datasets are utilized in this study. The first type consists of the county-level and town-level population data obtained from the Seventh National Population Census of China, which are considered as the dependent variable. The second type comprises the 100-m gridded auxiliary data, which are regarded as the independent variables (i.e., covariates). Finally, the third type is the inhabited data representing the areas of human activities (Baynes et al., 2022; Tu et al., 2022).

### 2.1 County-level and town-level census data

The seventh census data were collected at both county and town levels to generate population grids. The population count for 2848 counties across entire mainland China in 2020 was obtained from the seventh census data, as shown in Figure 1 (a). As the town-level census data have released for parts of towns, the population count of 15,564 towns within 1135 counties was also collected. The county-level and town-level population datasets were split into two subsets: a training set and a test set.

Due to the limited county-level samples, all were used for training. Taking the county as the basic sampling unit, we randomly selected 85% of town-level census data in Figure 1 (b) and they were combined with all county-level census data as the training set. That is to say, all counties in Figure 1 (a) and the towns in Figure 1 (b) were combined to train the base models and the stacking ensemble learning for estimating population grids. The remaining 15% of town-level census data were formed as the test set, as depicted in Figure 1 (c). The test samples were randomly distributed across China, indicating their feasibility for evaluating the fitted models. In this study, Hong Kong, Macao, and Taiwan were excluded due to the different conduction of the census.

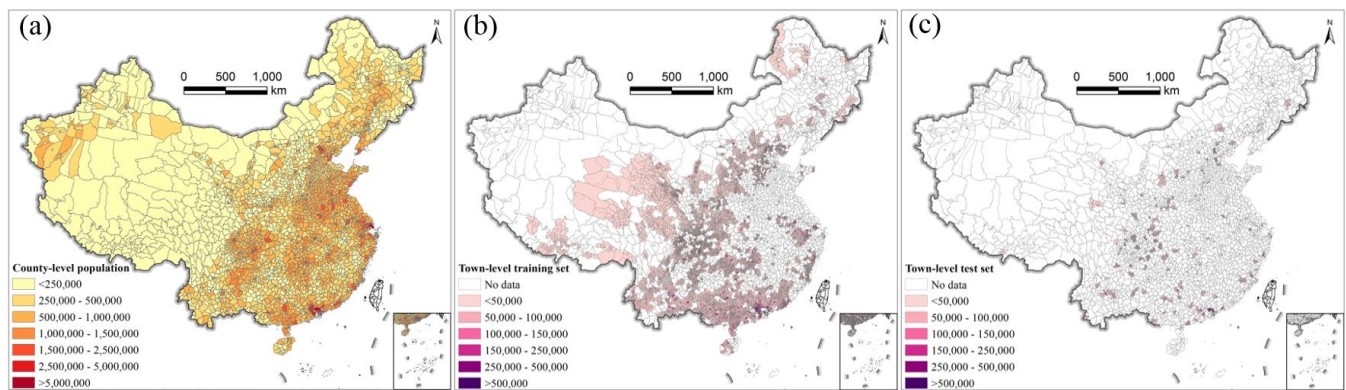

**Figure 1: The seventh census data of China. (a) The county-level census data, (b) The town-level census data for training, (c) The town-level census data for test.**

## 2.2 Gridded auxiliary data of population

Gridded auxiliary datasets play a pivotal role in estimating accurate gridded population distributions. In accordance with previous studies (Chen et al., 2016; Chen et al., 2020a; Cheng et al., 2020; Tu et al., 2022; Tatem, 2017; Ye et al., 2019; Zhao et al., 2020), we collected eight categories of 100-m gridded geospatial big data that are associated with population distribution in Figure 2.

The Tencent density user positioning data, sourced from China's largest social media company, Tencent, has been convincingly validated as a reliable proxy of human distribution (Chen et al., 2019; Tu et al., 2022; Chen et al., 2022a; Cheng et al., 2020). Real-time density images of Tencent user positions, captured at a resolution of 0.01° (https://heat.qq.com/), were collected every five minutes between January 1 and June 30 in 2019. These images were subsequently averaged to yield an aggregated Tencent user density image. A projection transformation was conducted and the bilinear resampling process was further implemented to convert it into a 100-m Tencent user density image, as illustrated in Figure 2 (a).

Points of interest (POIs) often represent important places of human activities and they are valuable for characterizing population distribution. We gathered an extensive dataset of over 60 million POIs from AutoNavi Maps (https://amap.com/), one of China's prominent online map platforms. Only POIs associated with human activities were used. These POIs were classified into ten main categories: restaurant, shopping, life service, working, education, medical facility, residence,

transportation, recreation, and others. The Point Density tool in ArcMap summarized the number of POIs within each 100-m grid and this information was used as the POI density in Figure 2 (b).

Human travel and activities are heavily reliant on road networks, and a higher density of roads often corresponds to increased human activity. The length of roads within each 100-m grid was computed by the Line Density tool in ArcMap using the road data acquired from the online map of AutoNavi Maps and it was considered as the road density in Figure 2 (c).

Note that the roads used in this study mainly included city roads, as well as provincial, county, and township-level roads, while excluding railways and expressways.

The night-time light (NTL) data are proficient in effectively characterizing nocturnal human activities, and they have been demonstrated as a significant indictor of human distribution (Elvidge et al., 2021). The annually composited Visible Infrared Imaging Radiometer Suite (VIIRS) NTL image in 2020 was acquired from the website:

https://eogdata.mines.edu/nighttime_light/annual/v20/. The original VIIRS NTL image, initially at a resolution of 500-m, was resampled by the bilinear algorithm to a 100-m NTL image as one covariate, as depicted in Figure 2 (d).

Taller buildings tend to accommodate a more population and there exists a strong correlation between building height and population distribution. The 10-m building height of China in 2020 was first collected (Wu et al., 2023). It was then aggregated to the 100-m building height covariate in Figure 2 (e).

The built area is the geographical space covered by both residential and non-residential buildings, serving as primary locations for human activities. It is also related to population distribution. The 10-m land cover data of China in 2020 released by Esri Inc. (https://www.arcgis.com/apps/instant/media/index.html?appid=fc92d38533d440078f17678ebc20e8e2) was first achieved and then the class of built area was extracted to calculate the percentage of built area within each 100-m grid in Figure 2 (f).

Digital elevation model (DEM) data are widely used in population downscaling, such as the WorldPop dataset (Tatem, 2017; Stevens et al., 2015) and the gridded population estimations in China (Cheng et al., 2020; Ye et al., 2019; Tu et al., 2022). In this context, the 30-m DEM data known as ALOS World 3D-30m (AW3D30) was procured from the official website: https://www.eorc.jaxa.jp/ALOS/en/aw3d30/data/index.htm. For analytical purposes, this dataset was resampled to 100-m DEM by the nearest algorithm in Figure 2 (g) and its 100-m slope was further calculated in Figure 2 (h).

Two location-related data, longitude and latitude were calculated for each 100-m grid to account for the geographical properties of the dependent variable and its covariates.

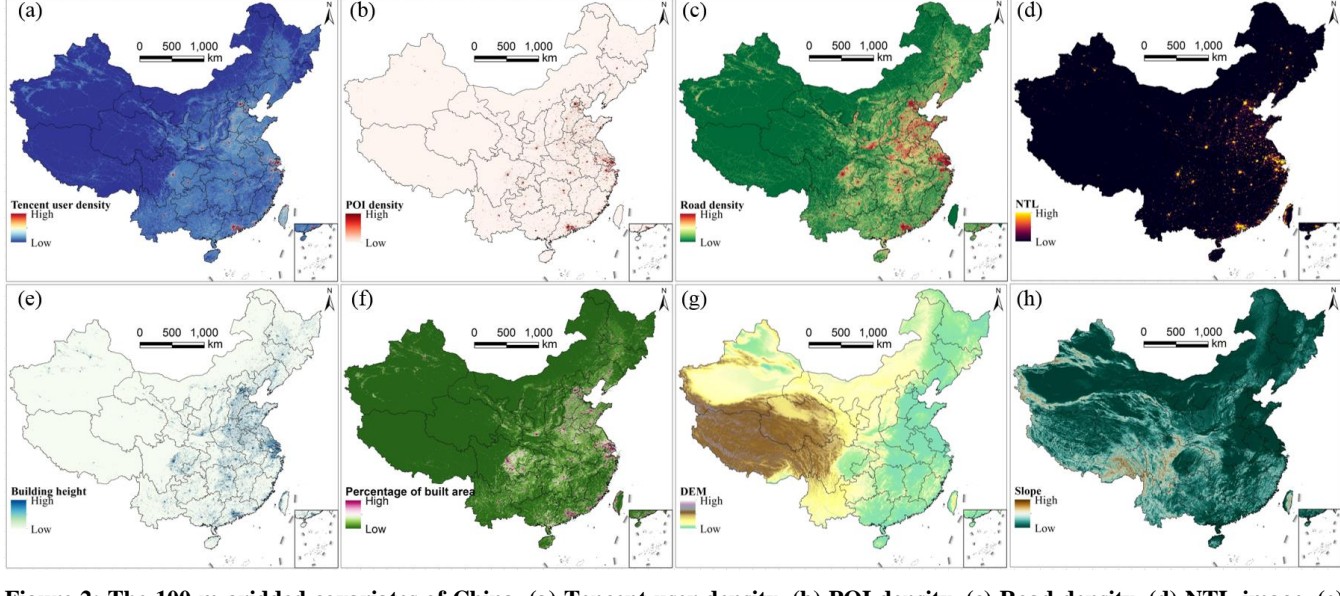

**Figure 2: The 100-m gridded covariates of China. (a) Tencent user density, (b) POI density, (c) Road density, (d) NTL image, (e) Building height, (f) Percentage of built area, (g) DEM, (h) Slope.**

Ten ultimate gridded covariates at a 100-m resolution were extracted from the eight categories of auxiliary data. They were employed in downscaling the seventh census data of mainland China to population grids, as illustrated in Table 1.

**Table 1: Covariates for the population downscaling of China.**

| Covariate | Description and source | Time |
|---|---|---|
| Tencent user density image | Tencent user positioning data | 2019 |
| POI density image | Number of POIs within each grid from the online map of AutoNavi Maps | 2020 |
| Road density image | Road length within each grid from the online map of AutoNavi Maps | 2020 |
| NTL image | VIIRS NTL data | 2020 |
| Building height image | Building height data | 2020 |
| Percentage of built area | The proportion of built area within each grid from Esri 10-m global land cover data | 2020 |
| DEM | ALOS World 3D-30m (AW3D30) data | - |
| Slope | | |
| Longitude image | Centroid of girds | - |
| Latitude image | | |

## 2.3 Inhabited area data

Recent studies on population downscaling have highlighted the effectiveness of excluding uninhabited areas to enhance the accuracy of gridded population estimates (Baynes et al., 2022; Tu et al., 2022). Usually, inhabited areas are identified as regions with human activities. Therefore, we generated inhabited areas using the gathered gridded geospatial big data in Figure 2. According to the inhabited area definition (Baynes et al., 2022; Tu et al., 2022), these areas should contain at least one none-zero human activity-related covariates. To achieve this, we employed the six covariates depicted in Figure 2 (a)-(f) to extract the inhabited areas of China, as illustrated in Figure 3. During the census data downscaling process, only grids falling within these inhabited areas were used to allocate the population counts.

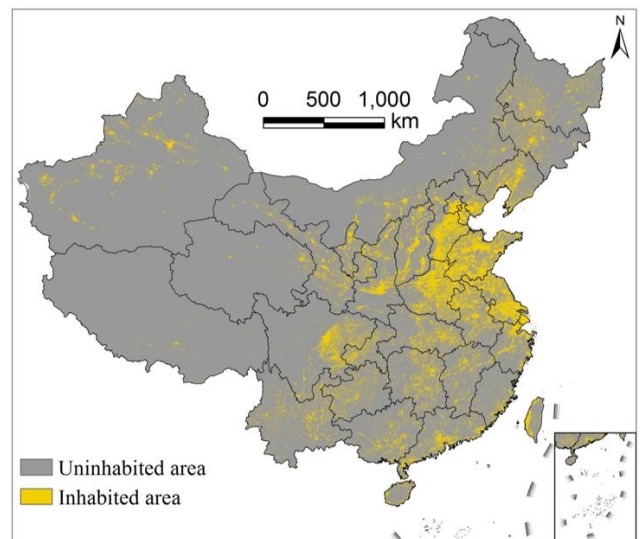

**Figure 3: The 100-m inhabited areas of China.**

## 3 Methodology

We devised a population downscaling approach by stacking ensemble learning (PopSE) to produce population grids. The framework outlines the use of the proposed PopSE for generating population grids in China, as illustrated in Figure 4. The framework contains seven main steps as follows.

(1) Collecting population data and covariates. The county-level and town-level population data were obtained from the Seventh National Population Census of China. These data were spatially linked to their respective administrative boundaries, as depicted in Figure 1. Ten gridded covariates were collected and processed at the 100-m resolution in Figure 2. To facilitate analysis, both the population data and covariates underwent a transformation to the Albers equal-area conic coordinate system.

(2) Extracting the inhabited area. Six 100-m covariates (i.e., Tencent user density, POI density, road density, NTL image, building height, and percentage of built area) were used to generate the gridded inhabited area. According to the definition of

inhabited area, this was calculated using the logic operations of the Map Algebra tool in ArcMap. If a grid contains at least one none-zero value in each of the six covariates, the grid is designated as the inhabited area.

(3) Calculating county-level and town-level population density and covariates. Compared to population count, population density is more suitable for comparing regions of different sizes and is frequently used as the dependent variable in population estimations (Stevens et al., 2015; Cheng et al., 2020; Ye et al., 2019). Therefore, the county-level and town-level population density was calculated through dividing the population count of each census unit by the inhabited area of its corresponding unit. The logarithm of county-level and town-level population density was used as dependent variable during the training of the proposed PopSE. Ten county-level and town-level covariates were aggregated separately from the 100-m gridded covariates. This aggregation for covariates ensured spatial alignment with the county-level and town-level population density. The aggregated covariates were utilized as independent variables in the modelling process.

(4) Building and training PopSE. PopSE is built based on stacking ensemble learning to combine individual algorithms to achieve better result than any individual algorithm. Population density and covariates at both county-level and town-level serve as inputs for training the proposed PopSE.

(5) Predicting gridded population density using the trained PopSE. Utilizing 100-m gridded covariates as inputs, the trained PopSE model was applied to predict the 100-m gridded population density for China.

(6) Converting gridded population density to gridded population data. To maintain the coherency between the population count of each census unit and the aggregated sum of population grids within the unit from the gridded population density, an adjustment was conducted on the gridded population density to generate the gridded population count.

(7) Assessing and comparing gridded population data. The accuracy of the estimated 100-m population grids was evaluated using town-level population test data. This assessment aimed to compare the town-level census data with the corresponding town-level population count aggregated from the estimated population grids. Additionally, the WorldPop data and the LandScan data in 2020 were further gathered to compare with the estimated population grids of China in 2020.

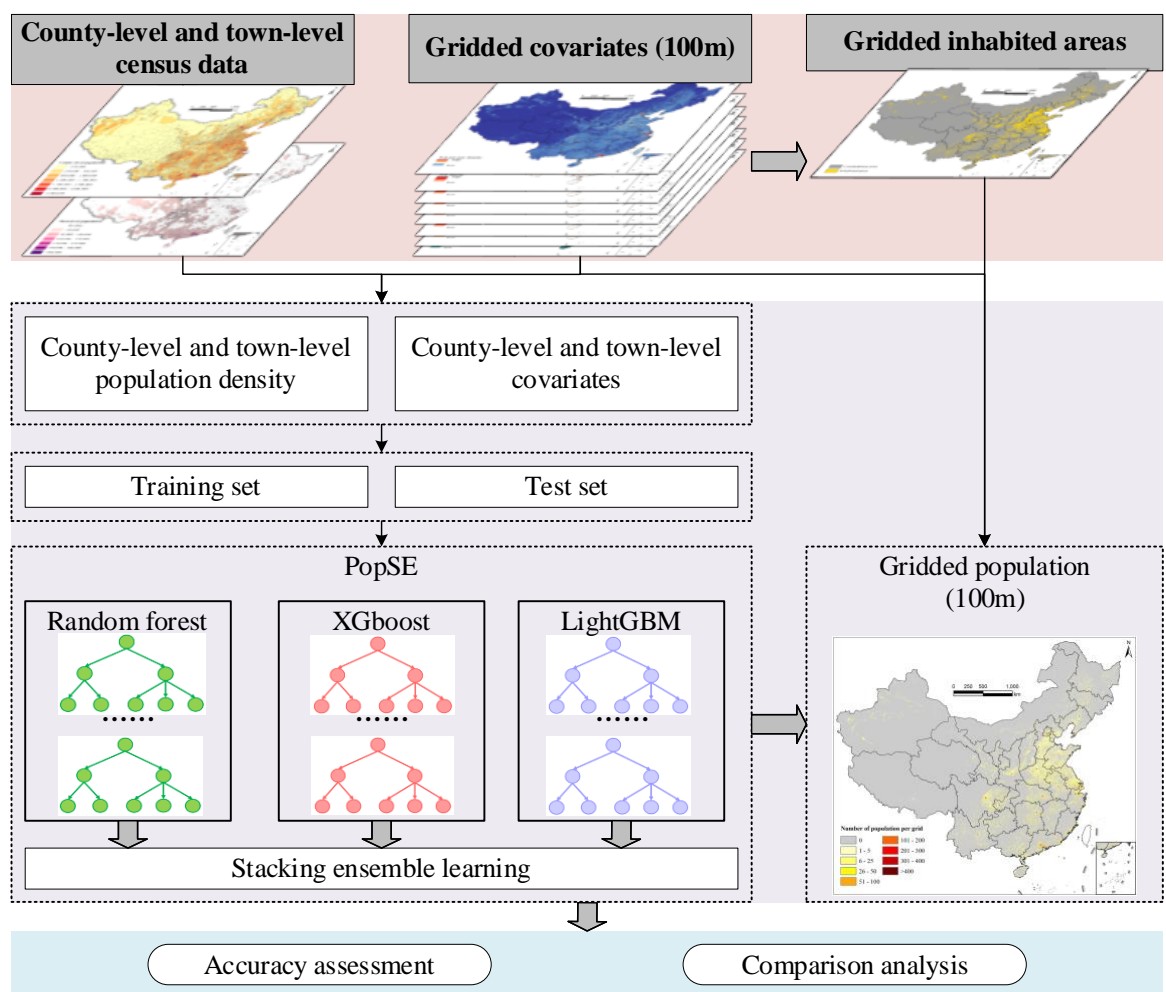

Figure 4: Framework of generating population grids for China.

### 3.1 Building PopSE

PopSE aims to leverage stacking ensemble learning to capitalize on the strengths of multiple individual machine learning algorithms for accurately characterizing the complex relationship between population distribution and its covariates. PopSE follows the principles of typical stacking ensemble learning and needs multiple base models and a metamodel, as shown in Figure 5. Three widely used algorithms—random forest, XGBoost, and LightGBM—were adopted as base models to construct PopSE. Random forest algorithm is a popular machine learning technique that trains different decision trees on various subsets of the training data to enhance accuracy and reduce variance and it is widely used in population downscaling (Cheng et al., 2020; Stevens et al., 2015; Zhao et al., 2020). XGBoost is a powerful machine learning algorithm that operates on an ensemble of decision trees using a gradient boosting framework and it is widely applied in various domains, including population and gross domestic product downscaling (Xu et al., 2023; Wu et al., 2020; Tu et al., 2022; Chen and Guestrin,

2016). LightGBM is a highly-efficient gradient boosting decision tree algorithm that achieves faster training speed and better accuracy through efficient histogram-based techniques and it is applied in diverse domains (Ke et al., 2017; Qiu et al., 2022; Xu et al., 2023; Chen et al., 2022b). PopSE inherits the common metamodel of linear regression in a standard stacking ensemble learning to amalgamate the predictions of the three base models.

**3.2 Training and testing**

With the population density and covariates as inputs, the base models and metamodel of the constructed PopSE can be fitted. The metamodel (i.e., linear regression) underwent training using cross-validation (CV) on the out-of-fold predictions from the base models. Before the PopSE training, we tuned the hyperparameters for random forest, XGBoost, and LightGBM using the grid search approach to achieve their optimum hyperparameters. During PopSE training, a 5-fold CV was used to

220 divide the training set into two parts, as illustrated in Figure 5. Four folds were utilized to fit each base model and the remaining fold was generated predictions from the fitted base models in Figure 5. This process iterated through the five folds. Finally, the metamodel was fitted using predictions from all base models. After the training, the test set was employed to evaluate the fitted PopSE. As shown in Figure 5, each base model was trained using four-fold training data and the fitted base model generated the prediction of the out-of-fold data. After five iterations, the five predictions from each base model

were combined to form new features for the meta model. The new feature set from the three base models was then used to fit the meta model, completing the stacking ensemble learning process.

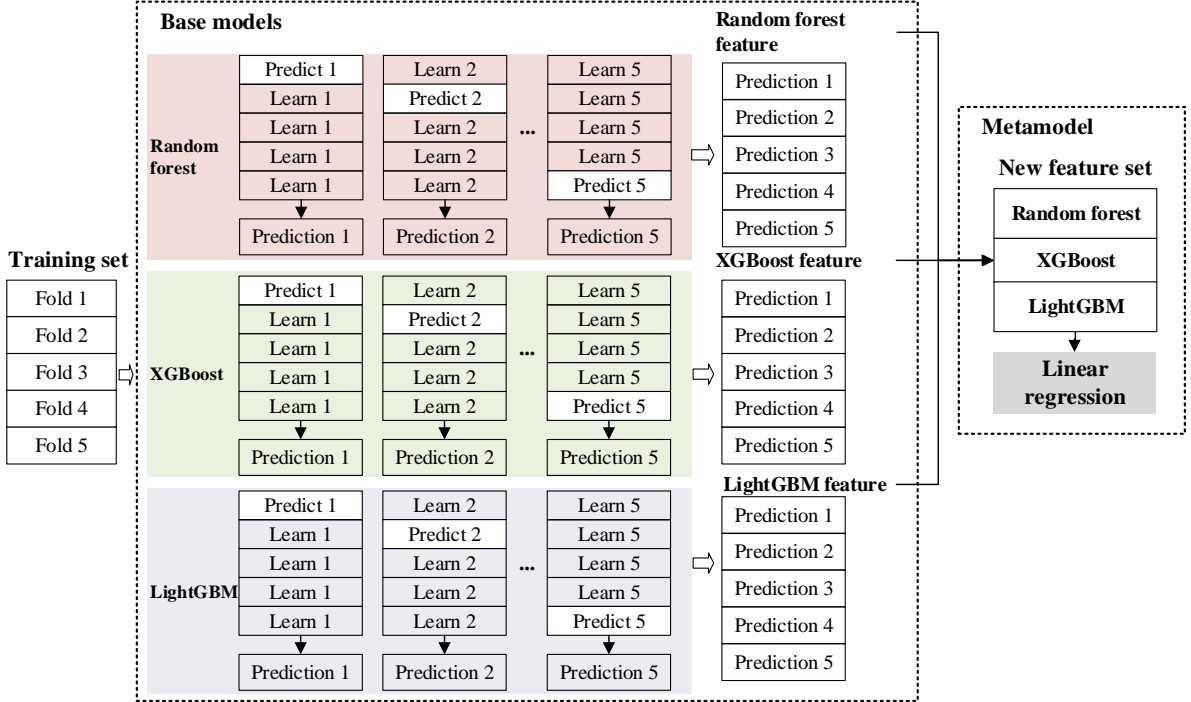

**Figure 5: Procedure of PopSE.**

### 3.3 Generating population grids

With the fitted PopSE and the gridded covariates, the 100-m gridded population density for China can be estimated by

$$D_j = e^{f(x_j)}, \tag{1}$$

where $D_j$ is the estimated population density for grid $j$, $f()$ is the fitted PopSE model, and $x_j$ is the vector of covariates for grid $j$.

To align the gridded population density with the population census count within each census unit, we adjusted the gridded

population density by multiplying the ratio of the population census count to the sum of estimated population grids within a census unit as

$$P_j = D_j \frac{C_i}{\sum_{j \in i} D_j}, \tag{2}$$

where $P_j$ is the estimated population count for grid $j$ within census unit $i$ and $C_i$ is the population count for census unit $i$.

### 3.4 Accuracy assessment

Due to the lack of population ground truth at a grid scale, the available town-level census data are the finest scale population data for accuracy assessment. Thus, the town-level population test set was adopted to evaluate the performance of the proposed PopSE and the existing gridded population products. The town-level population census counts in the test set were compared with the corresponding town-level population counts aggregated from gridded population data to compute performance metrics. Three metrics of the mean absolute error (MAE), the root mean square error (RMSE) and the

Coefficient of determination ($R^2$) were calculated for in the comparison. Their formulas are expressed as

$$MAE = \frac{1}{N} \sum_{k=1}^{N} |P_k - \widehat{P_k}|, \tag{3}$$

$$RMSE = \sqrt{\frac{1}{N} \sum_{k=1}^{N} (P_k - \widehat{P_k})^2}, \tag{4}$$

$$R^2 = 1 - \frac{\sum_{k=1}^{N}(P_k - \widehat{P_k})^2}{\sum_{k=1}^{N}(P_k - \bar{P})^2}, \tag{5}$$

where $P_k$ is the population census count for town $k$, $\widehat{P_k}$ is the estimated population count for town $k$, $\bar{P}$ is the mean of

estimated population count for all towns, and $N$ is the size of town-level population test set.

# 4 Results

## 4.1 Model evaluation

Both the base and PopSE models were initially fitted onto the training set using hyperparameters. Subsequently, predictions were generated from the test set. Table 2 presents the performance metrics for each model on the test set. Notably, the proposed PopSE exhibited superior performance over the three based models, as indicated by the highest $R^2$ (0.832) and the lowest RMSE (0.471) and MAE (0.304). In contrast, the random forest achieved the highest $R^2$ (0.826) and lowest RMSE (0.479) among the base models, while it recorded the highest MAE (0.317). The XGBoost achieved the worst performance among the base models, with the lowest $R^2$ (0.821) and the highest RMSE (0.486). For LightGBM, its performance metrics fell between those of the random forest and XGBoost. According to these metrics, the proposed PopSE performed the best on the test set, suggesting its potential to theoretically generate the most accurate gridded population dataset compared to the three base models.

**Table 2: Model performance metrics on population test set.**

| Model | $R^2$ | RMSE | MAE |
|---|---|---|---|
| Random forest | 0.826 | 0.479 | 0.317 |
| XGBoost | 0.821 | 0.486 | 0.314 |
| LightGBM | 0.825 | 0.481 | 0.311 |
| PopSE | 0.832 | 0.471 | 0.304 |

## 4.2 Gridded population map of China

The fitted PopSE utilized 100-m covariates to generate gridded population density and it was adjusted to the gridded population count using data from China's seventh census at both county-level and town-level. Figure 6 presents the gridded population map, derived from the proposed PopSE, at a spatial resolution of 100-m. Notably, numerous grids in the map exhibit zero population in uninhabited areas, mirroring the pattern observed in Figure 4. Areas with relatively high population grids are concentrated in southeastern China, including the Huanghuaihai Plain, the Sichuan Basin, the middle and lower reaches of the Yangtze River, and the Pearl River Basin. This distribution aligns with the spatial patterns observed in Figure 1. The gridded population map also reveals a hierarchical clustered distribution. The primary population hotspots in the first hierarchy distribute around urban agglomerations like the Beijing-Tianjin-Hebei region, the Pearl River Delta, and the Yangtze River Delta. In the second hierarchy, hotspots are predominantly found in provincial cities such as Chengdu, Chongqing, Xi'an, Zhengzhou, and Wuhan. The third hierarchy includes population hotspots in other city centers. Four representative zoom-in regions, namely Chengdu, Guangzhou, Beijing, and Nanjing, were selected for a detailed analysis of the spatial distribution of population grids. The examination of these four zoom-in regions revealed that city centers exhibit higher population grids compared to suburbs. In both city centers and suburbs of the four zoom-in regions, uninhabited areas

such as water surfaces and mountains show zero population. This spatial distribution of population grids aligns intuitively with the fundamental understanding of population patterns in China, suggesting the effectiveness of the proposed PopSE in visual.

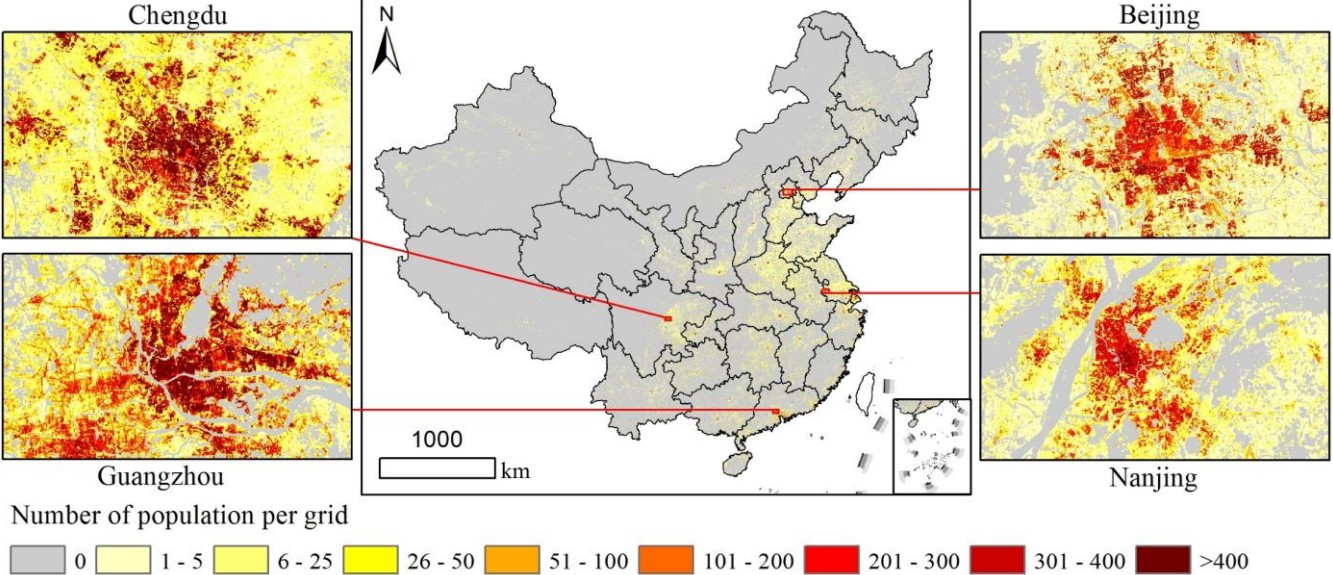

**Figure 6: The 100-m gridded population map of China in 2020 by PopSE.**

### 4.3 Accuracy assessment

The test set comprised 15% of town-level census data (i.e., 1931 towns). During accuracy assessment, it was deemed necessary to exclude this set in adjusting estimated population density grids to population count grids. Consequently, for accuracy assessment, the 100-m population count grids were adjusted solely using county-level census data from estimated population density grids. The metrics for the accuracy assessment of estimated population count grids by PopSE are presented in Figure 7. It can be found from Figure 7 that the $R^2$ reached a high value (0.8936), indicating the 100-m estimated population grids achieved a high accuracy at the town-level scale. The RMSE (22798) and MAE (10173) also imply a relatively low error in the estimated population grids. The coefficient (0.9904) of the fitted regression line in Figure 7 closely approximates 1, signifying a strong fit between the census population counts and the estimated population counts. This robustly demonstrates the effectiveness of the proposed PopSE.

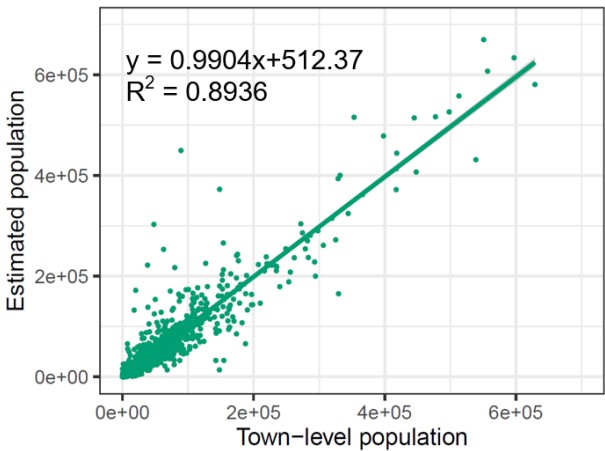

**Figure 7: Accuracy of the 100-m gridded population map of China by PopSE.**

## 5 Discussion

### 5.1 Comparison with two existing gridded population products

In this study, we collected two existing widely-used gridded population products to compare with the estimated gridded population dataset by the proposed PopSE. The 3 arc-seconds (~100-m) WorldPop dataset (i.e., Constrained individual countries 2020 UN adjusted) and the 30 arc-seconds (~1-km) LandScan dataset in 2020 were extracted and processed for the geographical region of China. The WorldPop dataset has the same spatial resolution to the estimated population grids, while the LandScan dataset is different from them. To ensure a consistent basis for comparison, the LandScan dataset was directly resampled to the spatial resolution of 3 arc-seconds (~100-m) with the Nearest Neighbor algorithm. It is evident from Figure 8 that areas with higher population grids are predominantly located in southeastern China, consistent with the patterns observed in Figure 6. The WorldPop product exhibits slightly more zero-population grids compared to the LandScan product, yet it closely aligns with estimated population dataset in Figure 6. Especially, the examination of four zoom-in regions reveals such detailed population distribution pattern. It means that LandScan product allocated more population to uninhabited areas like water surfaces and mountains. Focusing on the four regions, it can be further found that WorldPop shows fewer population grids exceeding 300 than LandScan. However, both products exhibit a lower number of high-value population grids, especially those exceeding 300, compared to the estimated population dataset in Figure 6. This suggests a potential underestimation of grid population counts in the two existing products. In addition, the four zoom-in regions show that LandScan is obviously coarser than both WorldPop and the estimated population datasets. The primary reason for this discrepancy is that the spatial resolution of LandScan product is one-tenth of other two gridded population products.

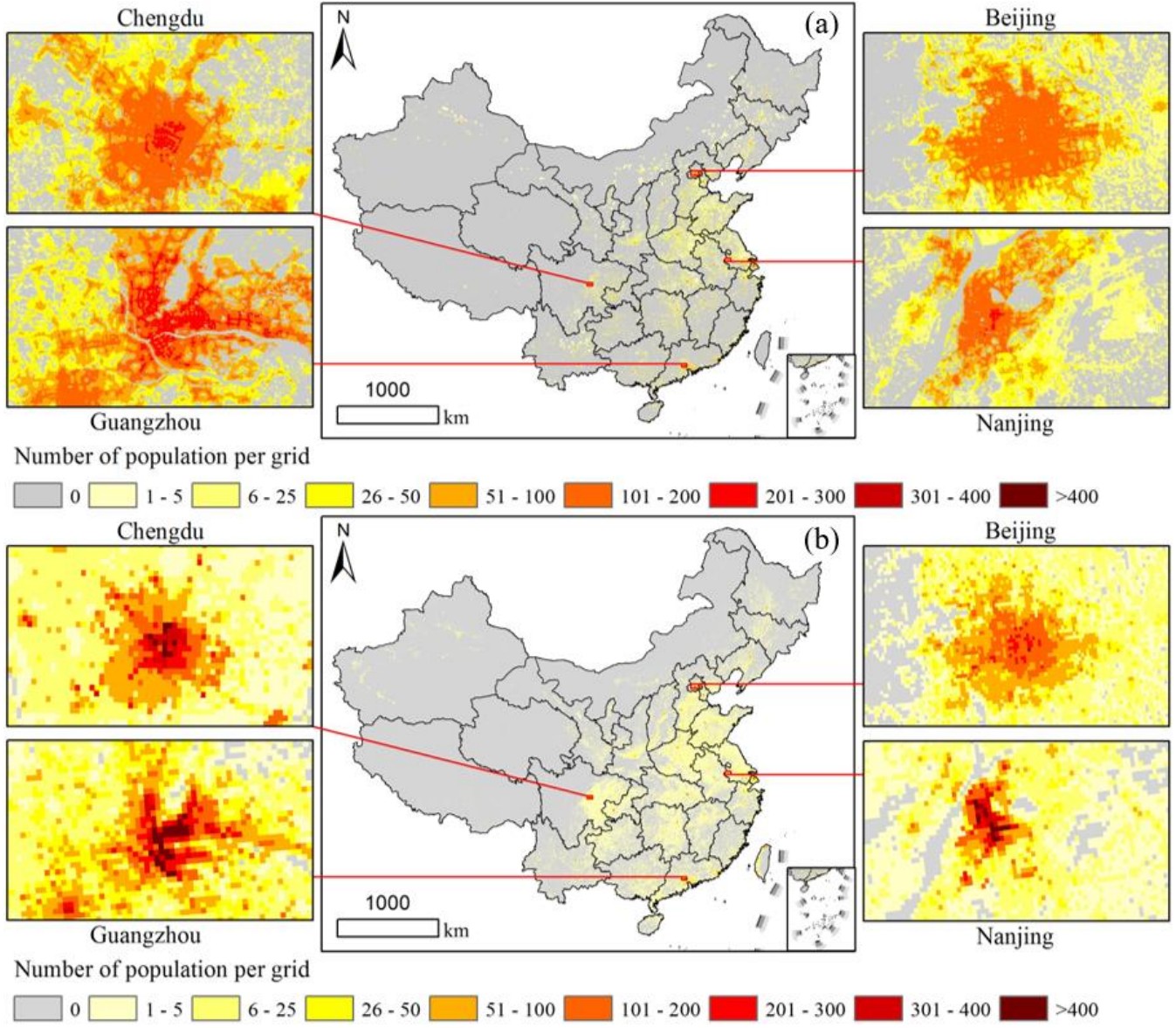

**Figure 8: Two existing gridded population maps of China in 2020. (a) WorldPop and (b) LandScan.**

The town-level census test set was also adopted to evaluate the accuracy of the two existing gridded population products. Figure 9 depicts the evaluated metrics for the two products. It can be seen from Figure 9 that the $R^2$ values for WorldPop and LandScan are 0.7427 and 0.7165, respectively, indicating a decrease of 0.1509 and 0.1771 compared to the estimated population product by PopSE in Figure 7. The RMSE of WorldPop and LandScan is separately 34315 and 36508, while the MAE of WorldPop and LandScan is 18366 and 17756, respectively. The RMSE and MAE for WorldPop and LandScan are

notably higher than those in Figure 7. Specifically, the RMSE for WorldPop and LandScan is respectively 1.5 and 1.6 times of that for the estimated product, and the MAE is separately 0.8 and 0.7 times higher. The coefficients of the fitted regression

line in Figure 9 are 0.8155 and 0.6136 for WorldPop and LandScan, respectively. These values are noticeably lower than 1 and also less than the coefficient in Figure 7. This quantitatively demonstrates an underestimation of gridded population counts for both WorldPop and LandScan.

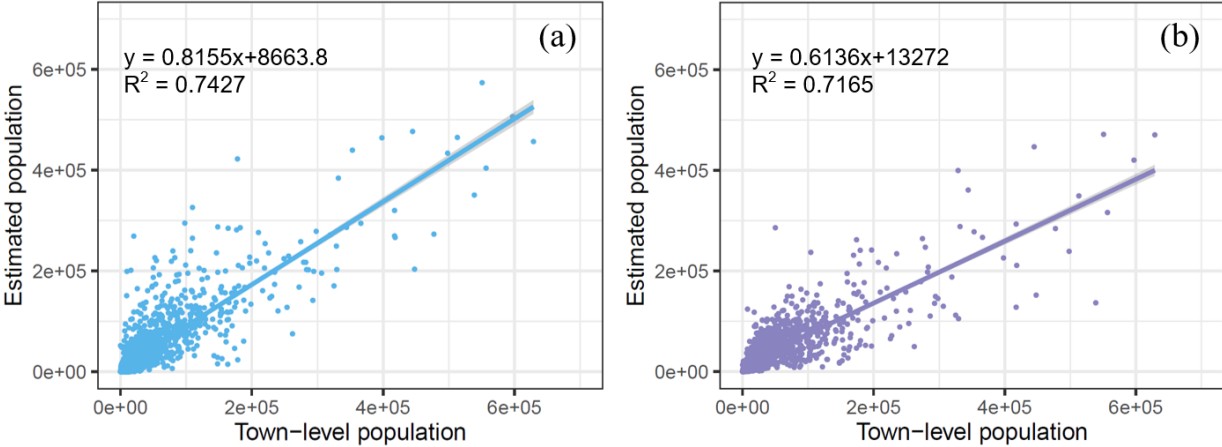

**Figure 9: Accuracy of two existing gridded population maps of China in 2020. (a) WorldPop and (b) LandScan.**

**5.2 Feature importance analysis in machine learning algorithms**

To investigate the influence of ten covariates on the fitted PopSE, the feature importance (i.e., weight) of covariates was obtained using the ELI5 Python package. It allows to show the feature importance of various machine learning algorithms,

including random forest, XGBoost, LightGB, and stacking ensemble learning. Figure 10 illustrates the feature importance of each covariate for the fitted PopSE and its three base models. Notably, POI density emerges as the most impactful on fitting PopSE and the three base models, with a significantly higher feature importance compared to the other nine covariates. Following closely are the four covariates of latitude, percentage of built area, NTL, and building height and they have similar importance level with relatively equal feature importance. Subsequently, the covariates of Tencent user density, slope, DEM,

and longitude exhibit comparable levels of feature importance in PopSE and the three base models. Road density has the lowest contribution to build PopSE with the smallest feature importance. Except building height, the feature importance of PopSE and its base models is comparable for each covariate in Figure 10.

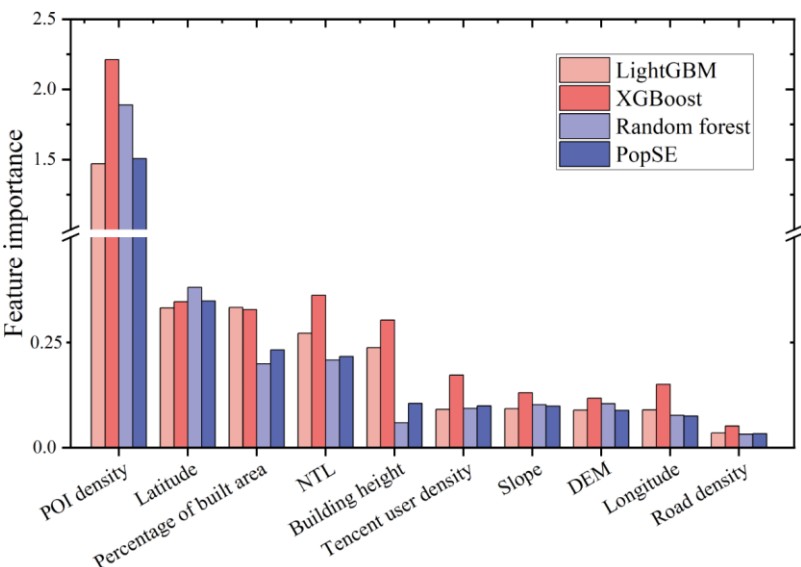

**Figure 10: Feature importance of the proposed PopSE and its three base models.**

### 5.3 Hyperparameter tuning

The three base models of PopSE incorporate multiple hyperparameters, which have substantial influence on the learning outcomes. Consequently, fine-tuning these hyperparameters is often imperative for achieving optimal performance. Our hyperparameter tuning employed the "grid search" technique by exploring a defined parameter search space. Specifically, we tuned the number of trees and the maximum depth of a tree for each base model. The search interval was set to 1 for both the number of trees and the maximum depth of a tree. The initial search space for each base model was determined through a combination of trial and error, along with empirical findings, as illustrated in Table 3. The optimal hyperparameter values were derived through cross validation utilizing the "grid search" technique, as shown in Table 3. Hyperparameters not listed in Table 3 retained default values from their respective Python packages.

**Table 3: Search space for hyperparameters in base models of PopSE.**

| Hyperparameter | Model | Search space | Optimal value |
|---|---|---|---|
| | Random forest | [90,120] | 105 |
| Number of trees | XGBoost | [100,130] | 105 |
| | LightGBM | [150,165] | 163 |
| | Random forest | [23,30] | 25 |
| Maximum depth of a tree | XGBoost | [5,15] | 6 |
| | LightGBM | [5,15] | 6 |

## 5.4 Advantages and Limitations

This study offers three main advantages compared to previous studies. First, we employed stacking ensemble learning to leverage the strengths of three popular machine learning algorithms (i.e., random forest, XGBoost, and LightGBM) and it can improve the overall performance and robustness of the gridded population estimation. Previous studies on estimating population grids often relied on a single machine learning algorithm and random forest was the most common choice (Cheng et al., 2020; Stevens et al., 2015; Ye et al., 2019). The results in Table 2 show that PopSE had better performance than the three base models, including random forest. Second, we utilized a variety of geospatial big datasets to predict accurate population grids and delineate the inhabited area to enhance population estimation. Previous studies often lacked either abundant covariates or detailed inhabited area data for national-scale population grid estimation (Cheng et al., 2020; Stevens et al., 2015; Ye et al., 2019; Chen et al., 2016; Chen et al., 2020c). Third, the proposed PopSE and geospatial big data were used to generate a 100-meter gridded population dataset from China's seventh census. This dataset outperformed two widely-used gridded population datasets (i.e., LandScan and WorldPop).

Although the proposed PopSE outperformed three base models on the test set and generated more accurate gridded population dataset for China than two existing products, this study still has its inherent limitations. The proposed PopSE shared similarities with many previous population downscaling methods and they assumed the scale invariance between the training set and the gridded covariates during training and prediction phases (Baynes et al., 2022; Chen et al., 2022a; Chen et al., 2020b; Cheng et al., 2020; Gao et al., 2021; Leyk et al., 2019; Qiu et al., 2022; Stevens et al., 2015; Wardrop et al., 2018; Ye et al., 2019; Zhao et al., 2020). There is often a shortage of gridded ground truth population data. Alternatively, census data collected in irregular administrative units were used as the ground truth for training population downscaling methods. Population downscaling methods, including the proposed PopSE fitted on census data, were typically executed on regular covariate grids to generate gridded population products. However, irregular census data has a spatial scale difference compared to the target regular population grids. This disparity may introduce uncertainty in the generation of population grids. At the same time, we employed only three widely used machine learning algorithms as base models, limiting the learning ability of the proposed PopSE.

Future work could benefit from incorporating more sophisticated and powerful algorithms into PopSE. Meanwhile, the use of high-quality covariates is crucial for generating accurate gridded population datasets. With the increasing availability of higher-resolution data, integrating more of these high-quality covariates (e.g., mobile phone signaling data, Weibo check-ins data, and house renting data) can further enhance the accuracy of gridded population datasets. In addition, the proposed method can be applied to other regions and time periods for generating gridded population datasets.

## 6 Data availability

The dataset of the 100-m gridded population counts for China in 2020 is stored in a GeoTIFF format and is freely available at https://figshare.com/s/d9dd5f9bb1a7f4fd3734 (Chen et al., 2024b).

## 7 Conclusions

In this study, we developed a novel population downscaling approach by leveraging stacking ensemble learning and geospatial big data. It aimed to employ stacking ensemble learning to combine the advantages of individual base models of random forest, XGBoost, and LightGBM. By integrating the predictions of these base models, the overall performance and robustness of gridded population estimation were enhanced compared to individual algorithms. Meanwhile, a variety of 100-m gridded geospatial big datasets were collected to delineate inhabited areas to specify and estimate population counts exclusively for China's seventh population census data. Experimental results have demonstrated that the proposed population downscaling approach outperformed individual base models and generated better gridded population dataset for China in 2020 than two existing gridded population products of WorldPop and LandScan in both quantitative and visual. Hence, the proposed population downscaling approach will be valuable option to generate population grids in other regions and the dataset described here will be useful for a wide range of applications like disaster and pollutant exposure assessment, resource and facility allocation, and more.

**Author contributions.** YC and YG designed the research and performed the analysis. YC and CX wrote the paper. XZ and YZ prepared the data and performed the analysis. YG edited and revised the paper. All authors contributed to and approved the final manuscript.

**Competing interests.** The contact author has declared that neither they nor their co-authors have any competing interests.

**Financial support.** This work was supported in part by the National Key R&D Program of China under Grant 2023YFC3006701 and in part by the National Natural Science Foundation of China under Grant 42071315.

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
