# Peer review of "A 100-m gridded population dataset of China's seventh census using ensemble learning and geospatial big data"

_Earth System Science Data, 2023_

## Author Comment (AC1)

**REVIEWER #1:**

Downscaling census data into population grids can address the limitations of census data with irregular units. This paper proposed a new population downscaling method using ensemble learning and geospatial big data. The method is adopted to generate a 100-m gridded population dataset of China's seventh census. The accuracy assessment on the generated population dataset shows that it has higher accuracy than the two existing datasets of WorldPop and LandScan. In general, the paper is well-written, clear, concise and complete in structure. I believe the generated dataset is important to a wide range of geoscience applications. Yet the following comments need to be addressed.

Response:

    Many thanks for your valuable suggestions and we deeply improved the manuscript according to your comments.

1. Line 100, what resample method is used for the variable?

Response:

    Thanks for your suggestion.

    We added the used method (i.e., bilinear resampling method) in the revised manuscript.

Added or revised contents:

    Line numbers in the revised manuscript: Line 113-115.

    [A projection transformation was conducted and the bilinear resampling process was further implemented to convert it into a 100-m Tencent user density image, as illustrated in Figure 2 (a).]

2. How to get the number of POIs and the road length for each grid in Section 2.2?

Response:

    Thanks for your suggestion.

    We used the Point Density tool in ArcMap to get the number of POIs within each grid. The Line Density tool in ArcMap was used to calculate the road length within each grid. We revised the descriptions.

Added or revised contents:

    Line numbers in the revised manuscript: Line 120-121 and 123-124.

    [The Point Density tool in ArcMap summarized the number of POIs within each 100-m grid and this information was used as the POI density in Figure 2 (b).]

[The length of roads within each 100-m grid was computed by the Line Density tool in ArcMap using the road data acquired from the online map of AutoNavi Maps and it was considered as the road density in Figure 2 (c).]

3. Although the authors provided the downscaling procedure (Figure 4), it's still unclear why do the 'population density' used instead of the direct population count.

Response:

Thanks for your suggestion.

It's similar to previous studies, we also used population density instead of population count. Because population density is more suitable for comparing regions of different sizes. We added the reason for this issue.

Added or revised contents:

Line numbers in the revised manuscript: Line 179-181.

[Compared to population count, population density is more suitable for comparing regions of different sizes and is frequently used as the dependent variable in population estimations (Stevens et al., 2015; Cheng et al., 2020; Ye et al., 2019).]

4. In Section 5.2, how do you get the feature importance for stacking? How about the three base models?

Response:

Thanks for your suggestion.

We used the ELI5 Python package to get the feature importance of stacking ensemble learning and we added the description for this issue. For the feature importance of the three base models, we revised Figure 10 and added their feature importance in Figure 10. Meanwhile, we also added their comparisons.

Added or revised contents:

Line numbers in the revised manuscript: Line 328-337.

[To investigate the influence of ten covariates on the fitted PopSE, the feature importance (i.e., weight) of covariates was obtained using the ELI5 Python package. It allows to show the feature importance of various machine learning algorithms, including random forest, XGBoost, LightGB, and stacking ensemble learning. Figure 10 illustrates the feature importance of each covariate for the fitted PopSE and its three base models. Notably, POI density emerges as the most impactful on fitting PopSE and the three base models, with a significantly higher feature importance compared to the other nine covariates. Following closely are the four covariates of latitude, percentage of built area, NTL, and building height and they have similar importance level with relatively equal feature importance. Subsequently, the covariates of Tencent user density, slope, DEM, and longitude exhibit comparable levels of feature importance in PopSE and the three base models. Road density has the lowest contribution to build PopSE with the smallest feature importance.

Except building height, the feature importance of PopSE and its base models is comparable for each covariate in Figure 10.]

[Figure]

Figure 10: Feature importance of the proposed PopSE and its three base models.

5. Section 5.3 presents the parameter selection, what's the search interval within the search space?

Response:

Thanks for your suggestion.

We added the search intervals for the hyperparameter tuning.

Added or revised contents:

Line numbers in the revised manuscript: Line 344-345.

[The search interval was set to 1 for both the number of trees and the maximum depth of a tree.]

6. Please add the caption of Figure 1c.

Response:

Thanks for your suggestion.

We added the caption for Figure 1c.

Added or revised contents:

[Figure]

Figure 1: The seventh census data of China. (a) The county-level census data, (b) The town-level census data for training, (c) The town-level census data for test.

7. Please enlarge the font size of Figure 1-3.

Response:

Thanks for your suggestion.

We revised the three figures.

Added or revised contents:

[Figure]

Figure 1: The seventh census data of China. (a) The county-level census data, (b) The town-level census data for training, (c) The town-level census data for test.

[Figure]

Figure 2: The 100-m gridded covariates of China. (a) Tencent user density, (b) POI density, (c) Road density, (d) NTL image, (e) Building height, (f) Percentage of built area, (g) DEM, (h) Slope.

[Figure]

Figure 3: The 100-m inhabited areas of China.

---

## Author Comment (AC2)

**REVIEWER #2:**

The authors develop a novel population downscaling approach that apply stacking ensemble learning and geospatial big data to produce 100-m population grids data. The RF, XGBoost, and LightGBM are combined to stack ensemble learning. Results show high accuracy and the proposed downscaling approach indeed provide a valuable option for producing gridded population datasets. However, there are still some problems to solve.

Response:

Many thanks for your valuable suggestions and we deeply improved the manuscript according to your comments.

1. The Introduction is well written. However, the author still need to add some research progress about the population estimation. The author should also add more about the machine learning method and ensemble learning method, and their comparison.

Response:

Thanks for your suggestion.

We added recent research progress about the population estimation and the comparison between machine learning and ensemble learning.

Added or revised contents:

Line numbers in the revised manuscript: Line 54-56 and 69-75.

[Apart from estimating population grids at the national scale in China, recent similar efforts have been made for individual cities and provinces (Guo et al., 2023a; Yang et al., 2023; Liu et al., 2023). Additionally, several global gridded population datasets are available for China, including LandScan and WorldPop (Bright and Coleman, 2000; Tatem, 2017).]

[Ensemble learning is a technique in machine learning and focuses on combining multiple algorithms to improve the overall performance and robustness of predictions (Dong et al., 2020). Stacking learning is considered to be one of the most effective ensemble learning techniques due to its utilization of a training mechanism to merge the predictions of individual machine learning algorithms (Costache and Bui, 2019). Ensemble learning has been widely recognized its merits in various applications (Dong et al., 2020; Wu et al., 2021; Xu et al., 2023). For instance, Yao et al. (2022) demonstrated the stacking ensemble learning outperformed typical individual machine learning algorithms in evaluating flash flood susceptibility.]

2. In line 110, the authors used the NTL data, and resample it into 100m. The authors should clarify the method for resampling.

Response:

Thanks for your suggestion.

We added the used method (i.e., bilinear resampling method) in the revised manuscript.

Added or revised contents:

Line numbers in the revised manuscript: Line 113-115 and 130-131.

[A projection transformation was conducted and the bilinear resampling process was further implemented to convert it into a 100-m Tencent user density image, as illustrated in Figure 2 (a).]

[The original VIIRS NTL image, initially at a resolution of 500-m, was resampled by the bilinear algorithm to a 100-m NTL image as one covariate, as depicted in Figure 2 (d).]

3. There are some expressions that should be more rigorous, such as "100 grid" in line 105 should be "100m grid"

Response:

Thanks for your suggestion.

We revised these typos.

Added or revised contents:

Line numbers in the revised manuscript: Line 120.

4. The authors used the DEM and Slope features from ALOS data for the population downscaling. However, did DEM and Slope features have strong correlation with population density. In west of China, such as Chongqing, population also distributed in areas with large elevation and slope.

Response:

Thanks for your suggestion.

We followed previous studies on global and national population grid estimations and DEM and slope features may be associated with population distribution. It is indeed that the two features do not have strong correlation with population distribution according to the results of feature importance for the two features. Even though, machine learning algorithms are different from linear regression and they can automatically reduce the impact of independent variables that are not strongly correlated to dependent variable. We also added the descriptions for this issue.

Added or revised contents:

Line numbers in the revised manuscript: Line 140-141.

[Digital elevation model (DEM) data are widely used in population downscaling, such as the WorldPop dataset (Tatem, 2017; Stevens et al., 2015) and the gridded population estimations in China (Cheng et al., 2020; Ye et al., 2019; Tu et al., 2022).]

5.    The authors applied the inhabited area data to enhance the accuracy of gridded population estimates. The authors should clarify the data and method to extract the inhabited area. There are also some errors for this step, and how to minimize this problem.

Response:

Thanks for your suggestion.

We added the calculation of the inhabited area.

Added or revised contents:

Line numbers in the revised manuscript: Line 160-161 and 175-178.

[According to the inhabited area definition (Baynes et al., 2022; Tu et al., 2022), these areas should contain at least one none-zero human activity-related covariates.]

[(2) Extracting the inhabited area. Six 100-m covariates (i.e., Tencent user density, POI density, road density, NTL image, building height, and percentage of built area) were used to generate the gridded inhabited area. According to the definition of inhabited area, this was calculated using the logic operations of the Map Algebra tool in ArcMap. If a grid contains at least one none-zero value in each of the six covariates, the grid is designated as the inhabited area.]

6.    The training data and test data are essential for the population estimation. The authors should introduce more about this.

Response:

Thanks for your suggestion.

The training and tests were described in Section 2.1. According to your comment, we also added the description for training and test sets.

Added or revised contents:

Line numbers in the revised manuscript: Line 96-99.

[That is to say, all counties in Figure 1 (a) and the towns in Figure 1 (b) were combined to train the base models and the stacking ensemble learning for estimating population grids. The remaining 15% of town-level census data were formed as the test set, as depicted in Figure 1 (c). The test samples were randomly distributed across China, indicating their feasibility for evaluating the fitted models.]

7.    The stacking ensemble learning is the important part of this work. What is the technical innovation of proposed method when compared with other method. What is the technical reference for similar studies.

Response:

Thanks for your suggestion.

We added the technical innovation and our advantages in the revised manuscript.

Added or revised contents:

Line numbers in the revised manuscript: Line 352-357.

[This study offers three main advantages compared to previous studies. First, we employed stacking ensemble learning to leverage the strengths of three popular machine learning algorithms (i.e., random forest, XGBoost, and LightGBM) and it can improve the overall performance and robustness of the gridded population estimation. Previous studies on estimating population grids often relied on a single machine learning algorithm and random forest was the most common choice (Cheng et al., 2020; Stevens et al., 2015; Ye et al., 2019). The results in Table 2 show that PopSE had better performance than the three base models, including random forest.]

8.    The discussion part should be deeper about the methods, result comparison, advantages and disadvantages, model transferability, validation, future research prospect and etc.

Response:

Thanks for your suggestion.

We improved the discussion parts to add the advantages and future works for this work. Meanwhile, Section 5.2 was improved in depth.

Added or revised contents:

Line numbers in the revised manuscript: Line 352-362 and 375-379.

[This study offers three main advantages compared to previous studies. First, we employed stacking ensemble learning to leverage the strengths of three popular machine learning algorithms (i.e., random forest, XGBoost, and LightGBM) and it can improve the overall performance and robustness of the gridded population estimation. Previous studies on estimating population grids often relied on a single machine learning algorithm and random forest was the most common choice (Cheng et al., 2020; Stevens et al., 2015; Ye et al., 2019). The results in Table 2 show that PopSE had better performance than the three base models, including random forest. Second, we utilized a variety of geospatial big datasets to predict accurate population grids and delineate the inhabited area to enhance population estimation. Previous studies often lacked either abundant covariates or detailed inhabited area data for national-scale population grid estimation (Cheng et al., 2020; Stevens et al., 2015; Ye et al., 2019; Chen et al., 2016; Chen et al., 2020c). Third, the proposed PopSE and geospatial big data were used to generate a 100-meter gridded population dataset from China's seventh census. This dataset outperformed two widely-used gridded population datasets (i.e., LandScan and WorldPop).]

[Future work could benefit from incorporating more sophisticated and powerful algorithms into PopSE. Meanwhile, the use of high-quality covariates is crucial for generating accurate gridded population datasets. With the increasing availability of higher-resolution data, integrating more of these high-quality covariates can further enhance the accuracy of gridded population datasets. In

addition, the proposed method can be applied to other regions and time periods for generating gridded population datasets.]

9.    The figure should be improved. For figure4, the text is too large. Figure 5 should clarify more about the ensemble learning. Figure 7 should be more standard and beautiful. Figure 8 should show the comparison between existing products and estimated results, not only the existing products. The unit of y-axis in Figure 10 should be clarified.

Response:

Thanks for your suggestion.

We revised the text size in Figure 4. We added description for Figure 5 to clarify the PopSE. We revised Figure 7 and Figure 9 to be standard. The result of the proposed PopSE is shown in Figure 6 and existing two products are shown in Figure 8 for the comparison. There is no unit of y-axis in Figure 10 as the value in y-axis is the feature importance.

Added or revised contents:

Line numbers in the revised manuscript: Line 223-226.

[As shown in Figure 5, each base model was trained using four-fold training data and the fitted base model generated the prediction of the out-of-fold data. After five iterations, the five predictions from each base model were combined to form new features for the meta model. The new feature set from the three base models was then used to fit the meta model, completing the stacking ensemble learning process.]

[Figure]

Figure 4: Framework of generating population grids for China.

[Figure]

Figure 5: Procedure of PopSE.

[Figure]

Figure 7: Accuracy of the 100-m gridded population map of China by PopSE.

[Figure]

Figure 9: Accuracy of two existing gridded population maps of China in 2020. (a) WorldPop and (b) LandScan.

10.     There are lots of good information and findings to highlight for the paper. What is the main contribution of this work from the perspective of technical problem. Please add accordingly in the abstract and conclusion parts.

Response:

Thanks for your suggestion.

We added the technical innovation in abstract and conclusion parts.

Added or revised contents:

Line numbers in the revised manuscript: Line 13-16 and 385-387.

[The proposed approach employs stacking ensemble learning to integrate the strengths of random forest, XGBoost, and LightGBM through fusing their predictions in a training mechanism and it

delineates the inhabited areas from geospatial big data to enhance the gridded population estimation.]

[It aimed to employ stacking ensemble learning to combine the advantages of individual base models of random forest, XGBoost, and LightGBM. By integrating the predictions of these base models, the overall performance and robustness of gridded population estimation were enhanced compared to individual algorithms.]

11.	The manuscript still exhibits minor issues in language expression, including spelling and grammatical errors. For example, some sentences in the introduction are overly complex, and the transitions between different viewpoints could be made smoother.

Response:

Thanks for your suggestion.

We checked the manuscript thoroughly and we revised the errors.

---

## Author Comment (AC3)

**CC #1:**

1. Tencent density user positioning data belongs to instantaneous data, showing significant differences during the day and night, and even between different time periods. What was the filming time for this paper? However, no matter which time period Tencent density user positioning data is available, it is not suitable for census data and is more suitable for using survey data to predict population density.

Response:

Thanks for your suggestion.

Although Tencent density user positioning data is the instantaneous data, the average of Tencent density user positioning data during a long period has been proved to be a reliable proxy for the human distribution in many previous studies. Therefore, we used the average image of Tencent density user positioning data from January 1 and June 30 in 2019. During this period, we obtained the instantaneous Tencent density user positioning data every five minutes. The description is shown in Line 109-113 in the revised manuscript.

2. Similarly, please explain how to obtain the feature importance by stacking ensemble learning . Is the ranking of feature importance applicable to the population distribution of all regions in China due to its complex landforms?

Response:

Thanks for your suggestion.

We used the ELI5 Python package to get the feature importance of stacking ensemble learning and we added the description for this issue. The training set included all counties and most towns across the entire China and the models were fitted by the training set. Therefore, the fitted models and the feature importance are applicable to the entire China.

Added or revised contents:

Line numbers in the revised manuscript: Line 328-330.

[To investigate the influence of ten covariates on the fitted PopSE, the feature importance (i.e., weight) of covariates was obtained using the ELI5 Python package. It allows to show the feature importance of various machine learning algorithms, including random forest, XGBoost, LightGB, and stacking ensemble learning.]

3. The article claims to have used a extensive dataset of 60 million POIs. What is the specific POI category for the application? The impact of POI categories on the population varies. They should be selected and weighted.

Response:

Thanks for your suggestion.

We added the categories for used POIs.

Added or revised contents:

Line numbers in the revised manuscript: Line 118-120.

[Only POIs associated with human activities were used. These POIs were classified into ten main categories: restaurant, shopping, life service, working, education, medical facility, residence, transportation, recreation, and others.]

4. Road data should also specify the category of use. For example, Expressways are not suitable for introduction, while Expressway toll stations are suitable for introduction.

Response:

Thanks for your suggestion.

The used road data exclude the expressways and railways and we added the categories for used road data.

Added or revised contents:

Line numbers in the revised manuscript: Line 126-126.

[Note that the roads used in this study mainly included city roads, as well as provincial, county, and township-level roads, while excluding railways and expressways.]

---

## Author Response (AR2)

**REVIEWER'S COMMENTS AND OUR RESPONSES**

We thank the editor and reviewers for their constructive comments. Hereafter we copy the comments and explain how we addressed these comments in the revised manuscript.

**REVIEWER #1:**

The authors have addressed my comments.

Response:

    Many thanks for your valuable comments for improving the manuscript.

**REVIEWER #2:**

I appreciate the author's efforts to address the issues I raised and to improve the quality of the article. The introduction part is well organized and some unclear statements are revised. Some of the figures are improved to make them more standard and clearer. I believe this gridded population dataset with high accuracy can be the important basis for the related studies. However, there are still some minor suggestions that should be considered before publications.

Response:

    Many thanks for your valuable suggestions and we further improved the manuscript according to your comments.

(1) The points in the scatter plot should be smaller.

Response:

    Thanks for your suggestion.

    We revised Figure 7 and Figure 9 with smaller points.

(2) Add some references on gridded population studies of China.

Response:

    Thanks for your suggestion.

    We added six related references on gridded population studies for China.

Added or revised contents:

    Line numbers in the revised manuscript: Line 55-56.

(3) Add the potentials of some other kinds of geographical big data in the production of large-scale gridded population datasets, such as mobile phone signaling data, Weibo check-ins data, and house renting data.

Response:

Thanks for your suggestion.

We added the potential data for the production of gridded population datasets.

Added or revised contents:

Line numbers in the revised manuscript: Line 377-378.

**THE EDITORIAL SUPPORT TEAM**

Figures 1, 2, 3, 4, 6, 8 may contain a territory that is disputed according to the United Nations. If and when the manuscript is accepted for final revised publication, you will be asked to choose one of the following options: (a) you could remove the disputed territory from the maps and submit new figure files, or (b) we could add a statement that some figures contain disputed territories.

Response:

Many thanks for your comments.
We choose the option (b) by adding a statement that some figures contain disputed territories.